# GOAL-DRIVEN IMITATION LEARNING FROM OBSERVATION BY INFERRING GOAL PROXIMITY

## ABSTRACT

Humans can effectively learn to estimate how close they are to completing a desired task simply by watching others fulfill the task. To solve the task, they can then take actions towards states with higher estimated proximity to the goal. From this intuition, we propose a simple yet effective method for imitation learning that learns a goal proximity function from expert demonstrations and online agent experience, and then uses the learned proximity to provide a dense reward signal for training a policy to solve the task. By predicting task progress as the temporal distance to the goal, the goal proximity function improves generalization to unseen states over methods that aim to directly imitate expert behaviors. We demonstrate that our proposed method efficiently learns a set of goal-driven tasks from state-only demonstrations in navigation, robotic arm manipulation, and locomotion tasks.

## 1 INTRODUCTION

Humans are capable of effectively leveraging demonstrations from experts to solve a variety of tasks. Specifically, by watching others performing a task, we can learn to infer how close we are to completing the task, and then take actions towards states closer to the goal of the task. For example, after watching a few tutorial videos for chair assembly, we learn to infer how close an intermediate configuration of a chair is to completion. With the guidance of such a task progress estimate, we can efficiently learn to assemble the chair to progressively get closer to and eventually reach, the fully assembled chair.

Can machines likewise first learn an estimate of progress towards a goal from demonstrations and then use this estimate as guidance to move closer to and eventually reach the goal? Typical learning from demonstration (LfD) approaches (Pomerleau, 1989; Pathak et al., 2018; Finn et al., 2016) greedily imitate the expert policy and therefore suffer from accumulated errors causing a drift away from states seen in the demonstrations. On the other hand, adversarial imitation learning approaches (Ho & Ermon, 2016; Fu et al., 2018) encourage the agent to imitate expert trajectories with a learned reward that distinguishes agent and expert behaviors. However, such adversarially learned reward functions often overfit to the expert demonstrations and do not generalize to states not covered in the demonstrations (Zolna et al., 2019), leading to unsuccessful policy learning.

Inspired by how humans leverage demonstrations to measure progress and complete tasks, we devise an imitation learning from observation (LfO) method which learns a task progress estimator and uses the task progress estimate as a dense reward signal for training a policy as illustrated in Figure 1. To measure the progress of a goal-driven task, we define *goal proximity* as an estimate of temporal distance to the goal (i.e., the number of actions required to reach the goal). In contrast to prior adversarial imitation learning algorithms, by having additional supervision of task progress and learning to predict it, the goal proximity function can acquire more structured task-relevant information, and hence generalize better to unseen states and provide better reward signals.

However, the goal proximity function can still output inaccurate predictions on states not in demonstrations, which results in unstable policy training. To improve the accuracy of the goal proximity function, we continually update the proximity function with trajectories both from expert and agent. In addition, we penalize trajectories with the uncertainty of the goal proximity prediction, which prevents the policy from exploiting high proximity estimates with high uncertainty. As a result, by leveraging the agent experience and predicting the proximity function uncertainty, our method can achieve more efficient and stable policy learning.

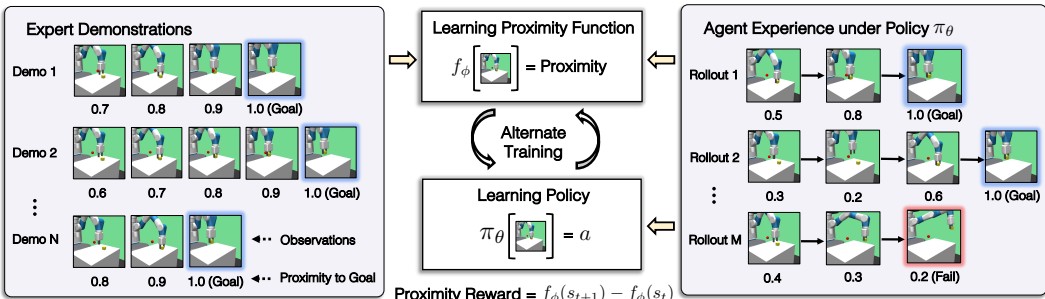

Figure 1: In goal-driven tasks, states on an expert trajectory have gradually increasing proximity toward the goal as the expert proceeds and fulfills a task. Inspired by this intuition, we propose to learn a proximity function $f_\phi$ from expert demonstrations and agent experience, which provides an estimate of temporal distance to the goal of a task. Then, using this learned proximity function, we train a policy $\pi_\theta$ to progressively move to states with higher proximity and eventually reach the goal to solve the task. We alternate these two learning phases to improve both the proximity function and the policy, leading to not only better learning efficiency but also superior performance.

The main contributions of this paper include (1) an algorithm for imitation from observation that uses estimated goal proximity to inform an agent of the task progress; (2) modeling uncertainty of goal proximity estimation to prevent exploiting uncertain predictions; and (3) a joint training algorithm of the goal proximity function and policy. We show that the policy learned with our proposed goal proximity function is more effective and generalizes better than the state-of-the-art LfO algorithms on various domains, such as navigation, robot manipulation, and locomotion. Moreover, our method demonstrates comparable results with GAIL (Ho & Ermon, 2016), which learns from expert actions.

## 2 RELATED WORK

Imitation learning (Schaal, 1997) aims to leverage expert demonstrations to acquire skills. While behavioral cloning (Pomerleau, 1989) is simple but effective with a large number of demonstrations, it suffers from compounding errors caused by the distributional drift (Ross et al., 2011). On the other hand, inverse reinforcement learning (Ng & Russell, 2000; Abbeel & Ng, 2004; Ziebart et al., 2008) estimates the underlying reward from demonstrations and learns a policy through reinforcement learning with this reward, which can better handle the compounding errors. Specifically, generative adversarial imitation learning (GAIL) (Ho & Ermon, 2016) and its variants (Fu et al., 2018; Kostrikov et al., 2020) shows improved demonstration efficiency by training a discriminator to distinguish expert and agent transitions and using the discriminator output as a reward for policy training.

While most imitation learning algorithms require expert actions, imitation learning from observation (LfO) approaches learn from state-only demonstrations. This enables the LfO methods to learn from diverse sources of demonstrations, such as human videos, demonstrations with different controllers, and other robots. To imitate demonstrations without expert actions, inverse dynamics models (Niekum et al., 2015; Torabi et al., 2018a; Pathak et al., 2018) or learned reward functions (Edwards et al., 2016; Sermanet et al., 2017; 2018; Liu et al., 2018; Lee et al., 2019a) can be used to train the policy. However, these methods require large amounts of data to train inverse dynamics models or representations. On the other hand, state-only adversarial imitation learning (Torabi et al., 2018b; Yang et al., 2019) can imitate an expert with few demonstrations, similar to GAIL. In addition to discriminating expert and agent trajectories, our method proposes to also estimate the proximity to the goal, which can provide more informed reward signals and generalize better.

Closely related works to our approach are reinforcement learning algorithms that learn a value function or proximity estimator from successful trajectories and use them as an auxiliary reward (Mataric, 1994; Edwards & Isbell, 2019; Lee et al., 2019b). While these value function and proximity estimator are similar to our proposed goal proximity function, these works require environment reward signals, and do not incorporate adversarial online training and uncertainty estimates.

Moreover, demonstrating the value of learning a proximity estimate for imitation learning, Angelov et al. (2020) utilizes the learned proximity to choose a proper sub-policy but does not train a policy

from the learned proximity. Similar to our method, Burke et al. (2020) proposes to learn a reward function using a ranking model and use it for policy optimization, demonstrating the advantage of using goal proximity as a reward for training a policy. However, they learn the proximity function from demonstrations alone and solely provide proximity as a reward. This hinders agent learning when the proximity function fails to generalize to agent experience, allowing the agent to exploit inaccurate proximity predictions for reward. By incorporating the online update, uncertainty estimates, and difference-based proximity reward, our method can robustly imitate state-only demonstrations to solve goal-driven tasks without access to the true environment reward.

## 3 METHOD

In this paper, we address the problem of learning from observations for goal-driven tasks. Adversarial imitation learning methods (Torabi et al., 2018b; Yang et al., 2019) suggest learning a reward function that penalizes an agent state transition off the expert trajectories. However, these learned reward functions often overfit to expert demonstrations and do not generalize to states which are not covered in the demonstrations, leading to unsuccessful policy learning.

To acquire a more structured and generalizable reward function from demonstrations, we propose to learn a *goal proximity function* that estimates proximity to the goal distribution in terms of temporal distance (i.e., number of actions required to reach the goal). Then, a policy learns to reach states with higher proximity (i.e., that are closer to the goal) predicted by the goal proximity function. Moreover, during policy training, we propose to measure the uncertainty of the goal proximity function which prevents the policy from exploiting over-optimistic proximity predictions and yielding undesired behaviors. In Section 3.2, we describe the goal proximity function in detail. Then, in Section 3.3, we elaborate how the policy is jointly trained with the goal proximity function.

### 3.1 PRELIMINARIES

We formulate our learning problem as a Markov decision process (Sutton, 1984) defined through a tuple $(\mathcal{S}, \mathcal{A}, R, P, \rho_0, \gamma)$ for the state space $\mathcal{S}$, action space $\mathcal{A}$, reward function $R(s, a)$, transition distribution $P(s'|s, a)$, initial state distribution $\rho_0$, and discounting factor $\gamma$. We define a policy $\pi(a|s)$ that maps from states to actions and correspondingly moves an agent to a new state according to the transition probabilities. The policy is trained to maximize the expected sum of discounted rewards, $\mathbb{E}_{(s,a)\sim\pi}\left[\sum_{t=0}^{T_i} \gamma^t R(s_t, a_t)\right]$, where $T_i$ represents the variable length of episode $i$.

In imitation learning, the learner receives a fixed set of expert demonstrations, $\mathcal{D}^e = \{\tau_1^e, \ldots, \tau_N^e\}$. In this paper, we specifically consider the learning from observation (LfO) setup where each demonstration $\tau_i^e$ is a sequence of states. Moreover, we assume that all expert demonstrations are successful; therefore, the final state of an expert trajectory reaches the task goal.

### 3.2 LEARNING GOAL PROXIMITY FUNCTION

In goal-driven tasks, an estimate of how close an agent is to the goal can be utilized as a direct learning signal. Therefore, instead of learning to discriminate agent and expert trajectories (Ho & Ermon, 2016; Torabi et al., 2018b), we propose a *goal proximity function*, $f : \mathcal{S} \to \mathbb{R}$, that learns how close states are to the goal distribution. Specifically, we define *goal proximity* as a proximity that is discounted based on its temporal distance to the goal (i.e., inversely proportional to the number of actions required to reach the goal). Note that the goal proximity function measures the temporal distance, not the spatial distance, between the current and goal states. Therefore, a single proximity value can entail all information about the task, goal, and any roadblocks.

In this paper, we define goal proximity of a state $s_t$ as the linearly discounted proximity $f(s_t) = 1 - \delta \cdot (T_i - t)$, where $\delta \in (0, 1)$ is a discounting factor and $T_i$ is the episode horizon. In this paper, we set $\delta$ to $1/H$ to evenly distribute the proximity between 0 and 1, where $H$ is the maximum task horizon. Note that we use the maximum episode length $H$, instead of the variable episode length $T_i$, to define a fixed $\delta$ for the temporal discounting to be consistent between episodes. We use mean squared error as the objective for training the goal proximity function $f_\phi$ parameterized by $\phi$:

$$\mathcal{L}_\phi = \mathbb{E}_{\tau_i^e \sim \mathcal{D}^e, s_t \sim \tau_i^e} \left[ f_\phi(s_t) - (1 - \delta \cdot (T_i - t)) \right]^2. \tag{1}$$

---

**Algorithm 1** Imitation learning with goal proximity function

---

**Require:** Expert demonstrations $\mathcal{D}^e = \{\tau_1^e, \ldots, \tau_N^e\}$
 1: Initialize weights of goal proximity function $f_\phi$ and policy $\pi_\theta$
 2: **for** $i = 0, 1, \ldots, M$ **do**
 3:     Sample expert demonstration $\tau^e \sim \mathcal{D}^e$                ▷ Offline proximity function training
 4:     Update goal proximity function $f_\phi$ with $\tau^e$ to minimize Equation 1
 5: **end for**
 6: **for** $i = 0, 1, \ldots, L$ **do**
 7:     Rollout trajectories $\tau_i = (s_0, \ldots, s_{T_i})$ with $\pi_\theta$                ▷ Policy training
 8:     Compute proximity reward $R_\phi(s_t, s_{t+1})$ for $(s_t, s_{t+1}) \sim \tau_i$ using Equation 5
 9:     Update $\pi_\theta$ using any RL algorithm
10:     Update $f_\phi$ with $\tau_i$ and $\tau^e \sim \mathcal{D}^e$ to minimize Equation 4
11: **end for**

---

There are alternative ways to represent and learn goal proximity, such as exponentially discounted proximity and ranking-based proximity (Brown et al., 2019). But, in our experiments, linearly discounted proximity consistently performed better than alternatives; therefore, the linearly discounted proximity is used throughout this paper (see Figure 5b and Figure 11).

By learning to predict the goal proximity, the goal proximity function not only learns to discriminate agent and expert trajectories (i.e., predict 0 proximity for an agent trajectory and positive proximity for an expert trajectory with Equation 4), but also acquires the task information about temporal progress entailed in the trajectories. From this additional supervision, the proximity function provides more informative learning signals to the policy and generalizes better to unseen states as empirically shown in Section 4.

## 3.3 TRAINING POLICY WITH PROXIMITY REWARD

In a goal-driven task, a policy $\pi_\theta$ aims to get close to and eventually reach the goal. We can formalize this objective as maximizing the goal proximity at the final state $f_\phi(s_{T_i})$, which can be used as a sparse proximity reward. In addition, to encourage the agent to make consistent progress towards the goal, we devise a dense proximity reward based on the increase in proximity, $f_\phi(s_{t+1}) - f_\phi(s_t)$, at every timestep. By combining the sparse and dense proximity rewards, our total proximity reward can be defined as

$$R_\phi(s_t, s_{t+1}) = \begin{cases} f_\phi(s_{t+1}) - f_\phi(s_t) & t \neq T_i - 1 \\ 2 \cdot f_\phi(s_{T_i}) - f_\phi(s_t) & t = T_i - 1 \end{cases}. \tag{2}$$

Given the proximity reward, the policy is trained to maximize the expected discounted return:

$$\mathbb{E}_{(s_0, \ldots, s_{T_i}) \sim \pi_\theta} \left[ \gamma^T f_\phi(s_{T_i}) + \sum_{t=0}^{T_i - 1} \gamma^t (f_\phi(s_{t+1}) - f_\phi(s_t)) \right]. \tag{3}$$

However, a policy trained with the proximity reward can sometimes perform undesired behaviors by exploiting over-optimistic proximity predictions on states not seen in the expert demonstrations. This becomes critical when the expert demonstrations are limited and cannot cover the state space sufficiently. To avoid inaccurate predictions leading an agent to undesired states, we propose to (1) fine-tune the goal proximity function with online agent experience to reduce optimistic proximity evaluations; and (2) penalize agent trajectories with high uncertainty in goal proximity predictions.

First, we set the target proximity of states in agent trajectories to 0, similar to adversarial imitation learning methods (Ho & Ermon, 2016), and train the proximity function with both expert demonstrations and agent experience by minimizing the following loss:

$$\mathcal{L}_\phi = \mathbb{E}_{\tau_i^e \sim \mathcal{D}^e, s_t \sim \tau_i^e} \left[ f_\phi(s_t) - (1 - \delta \cdot (T_i - t)) \right]^2 + \mathbb{E}_{\tau \sim \pi_\theta, s_t \sim \tau} \left[ f_\phi(s_t) \right]^2. \tag{4}$$

Although successful agent experience is also used as negative examples for training the proximity function, in practice, this is not problematic since the proximity function ideally converges to the average of expert and agent labels (e.g., $1/2 - \delta \cdot (T_i - t)/2$ for ours and 0.5 for GAIL). Early stopping and learning rate decay can be used to further ease this problem (Zolna et al., 2019). Also,

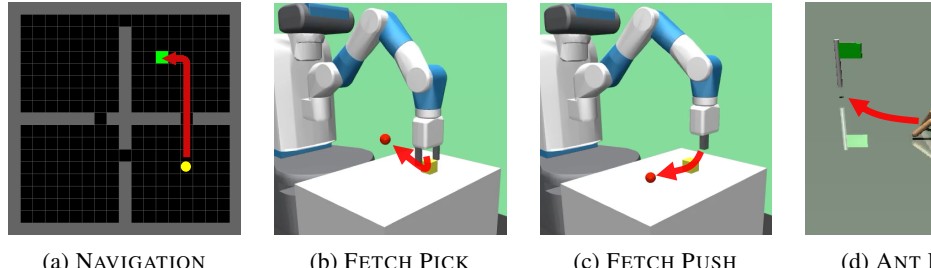

| (a) NAVIGATION | (b) FETCH PICK | (c) FETCH PUSH | (d) ANT REACH |

Figure 2: Four goal-driven tasks are used to evaluate our proposed method and the baselines. (a) The agent (yellow) must navigate across rooms to reach the goal (green). (b, c) The robotic arm is required to pick up or push the yellow block towards the goal (red). (d) A quadruped ant agent must walk towards the green flag.

the online training of the goal proximity function can lower the goal proximity estimate in a local optimum, which helps the policy escape from such local optima.

To alleviate the effect of inaccurate proximity estimation in policy training, we discourage the policy from visiting states with uncertain proximity estimates. Specifically, we model the uncertainty $U(s_t)$ as the disagreement in terms of standard deviation of an ensemble of proximity functions (Osband et al., 2016; Lakshminarayanan et al., 2017). Then we use the estimated uncertainty to penalize exploration of states with uncertain proximity estimates. The proximity estimate $f(s_t)$ is the average prediction of the ensemble. With the uncertainty penalty, the modified reward can be written as:

$$R_\phi(s_t, s_{t+1}) = \begin{cases} f_\phi(s_{t+1}) - f_\phi(s_t) - \lambda \cdot U(s_{t+1}) & t \neq T_i - 1 \\ 2 \cdot f_\phi(s_{T_i}) - f_\phi(s_t) - \lambda \cdot U(s_{T_i}) & t = T_i - 1 \end{cases}, \quad (5)$$

where $\lambda$ is a tunable hyperparameter to balance the proximity reward and uncertainty penalty. A larger $\lambda$ results in more conservative exploration outside the states covered by the expert demonstrations

In summary, we propose to learn a goal proximity function that estimates how close the current state is to the goal distribution, and train a policy to maximize the goal proximity while avoiding states with inaccurate proximity estimates using the uncertainty measure. We jointly train the proximity function and policy as described in Figure 1 and Algorithm 1.

## 4 EXPERIMENTS

In our experiments, we aim to answer the following questions: (1) How does our method's efficiency and final performance compare against prior work in imitation from observation and imitation learning with expert actions? (2) Does our method lead to policies that generalize better to states unseen in the demonstrations? (3) What factors contribute to the performance of our method? To answer these questions we consider diverse goal-driven tasks: navigation, robot manipulation, and locomotion.

To demonstrate the improved generalization capabilities of policies trained with the proximity reward, we benchmark our method under two different setups: expert demonstrations are collected from (1) only a fraction of the possible initial states (e.g., 25%, 50%, 75% coverage) and (2) initial states with smaller amounts of noise. (1) measures the ability of an agent to interpolate between states covered by the demonstrations while (2) evaluates extrapolating beyond the demonstrations to added noise during agent learning. In both setups, our method shows superior generalization capability and thus, achieves higher final rewards than LfO baselines. Moreover, our method achieves comparable results with LfD methods that use expert actions.

These generalization experiments serve to mimic the reality that expert demonstrations may be collected in a different setting from agent learning. For instance, due to the cost of demonstration collection, the demonstrations may poorly cover the state space. An agent would then have to learn in an area of the state space not covered by the demonstrations. We measure this in the experimental setup (1), where the demonstrations cover a fraction of the possible learner starting and goal states. Likewise, demonstrations may be collected in controlled circumstances with little environment noise. Then, an agent learning in an actual environment would encounter more noise than presented in

the demonstrations. We quantify this in the experimental setup (2), where less noise is applied to demonstration starting states.

## 4.1 BASELINES

We compare our method to the state-of-the-art prior works in both imitation learning from observations and standard imitation learning with actions, which are listed below:

- **BCO** (Torabi et al., 2018a) learns an inverse model from environment interaction to provide action labels in demonstrations for behavioral cloning.
- **GAIfO** (Torabi et al., 2018b) is a variant of GAIL (Ho & Ermon, 2016) which trains a discriminator to discriminate state transitions $(s, s')$ instead of state-action pairs $(s, a)$.
- **GAIfO-s**, as compared to in Yang et al. (2019), learns a discriminator based off only a single state, not a state transition as with GAIfO.
- **BC** (Pomerleau, 1989) fits a policy to the demonstration actions with supervised learning. This method requires expert action labels while our method does not.
- **GAIL** (Ho & Ermon, 2016) uses adversarial imitation learning with a discriminator trained on state-action pairs $(s, a)$. This method also uses actions whereas ours does not.

Also, we study several variations of our method to evaluate the importance of different design choices:

- **Ours (No Uncert)**: Removes the uncertainty penalty from the reward function.
- **Ours (No Online)**: Learns the proximity function offline from the demonstrations and does not refine it using agent experience during policy learning. This approach may fail as the proximity function will not learn outside of the demonstrations and thus provide a poor reward signal.
- **Ours (No Offline)**: Does not pre-train the proximity function. This should be less efficient than our method, which pre-trains the proximity function using the demonstrations.
- **Ours (Exp)**: Uses the exponentially discounted goal proximity $f(s_t) = \delta^{(T-t)}$.

## 4.2 EXPERIMENTAL SETUP

By default, our primary method uses the linearly discounted version of the proximity function as this empirically lead to the best results (see details in Figure 11) and set the discounting factor as $\delta = 1/H$, where $H$ is the task horizon length. For modeling uncertainty, we use an ensemble of size 5 across all tasks. For all tasks, we pre-train the proximity function for 5 epochs on the demonstrations. During online training (i.e., policy learning), we sample a batch of 128 elements from the expert and agent experience buffers. The mean and standard deviation of outputs from the ensemble networks are used as the proximity prediction and uncertainty, respectively.

The same network architecture is used for proximity function, discriminator (for the baselines), and policy. Details of the network architecture can be found in Section G.2. Any reinforcement learning algorithm can be used for policy optimization, but we choose to use PPO (Schulman et al., 2017) and the hyperparameters of PPO are tuned appropriately for each method and task (see Table 2). Each baseline implementation is verified against the results reported in its original paper. We train each task with 5 different random seeds and report mean and standard deviation divided by 2.

## 4.3 NAVIGATION

In the first set of experiments, we examine the NAVIGATION task between four rooms shown in Figure 2a. The purpose of this environment is to show the benefits of our method in a simple setting where we can easily visualize and verify the learned goal proximity function. The agent start and goal positions are randomly sampled and the agent has 100 steps to navigate to the goal. We provide 250 expert demonstrations obtained using a shortest path algorithm. During demonstration collection, we hold out 50% of the possible agent start and goal positions determined by uniform random sampling. In contrast, during agent learning and evaluation, start and goal positions are sampled from all possible positions.

As can be seen in Figure 3a, our method achieves near 100% success rate in 3M environment steps, while all GAIL variants fail to learn the task. Although BC and BCO could achieve the goal for about 60% and 30% cases respectively, they show limited generalization to unseen configurations. This

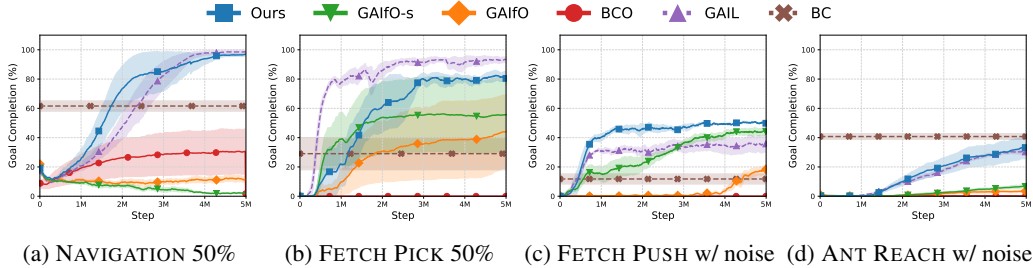

(a) NAVIGATION 50%    (b) FETCH PICK 50%    (c) FETCH PUSH w/ noise    (d) ANT REACH w/ noise

Figure 3: Goal completion rates of our method and baselines. The agent must generalize to a wider state distribution than seen in the expert demonstrations. Demonstrations in (a,b) cover only 50% of states and in (c,d) are generated with less noise. Note that GAIL and BC (dashed lines) use expert actions whereas all other methods, including ours, learn from observations only. Our method learns more stably, faster and achieves higher goal completion rates than baseline LfO algorithms. Moreover, our method outperforms BC and GAIL in NAVIGATION and FETCH PUSH, and achieves comparable results in all other tasks.

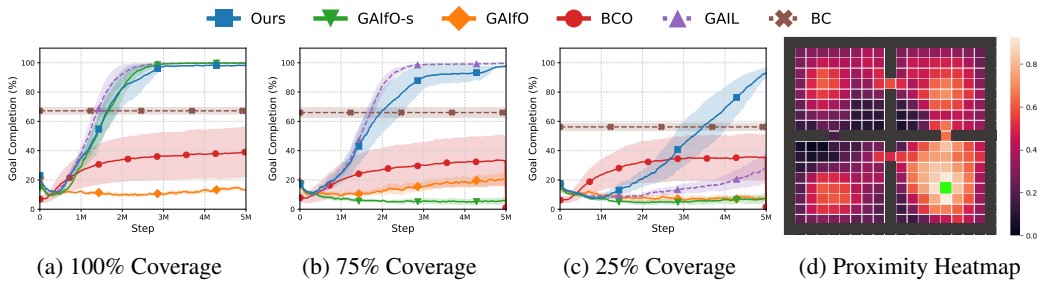

(a) 100% Coverage    (b) 75% Coverage    (c) 25% Coverage    (d) Proximity Heatmap

Figure 4: Analyzing the effect of improved generalization as the cause for performance increase in our method. (a) performance with no generalization required (i.e., same start and goal distribution for demonstrations and online learning). (b, c) performance with increasing difference between start and goal distributions of demonstrations and online learning. (d) visualization of the learned proximity function for a fixed goal (green). The proximity function was evaluated for every state on the grid; lighter cells correspond to states which higher estimated proximity to the goal.

result proves that learning with the goal proximity reward is effective and the learned goal proximity function generalizes well to unseen configurations.

To verify whether the proximity function learns meaningful information of proximity to the goal, we visualize the proximity from all agent positions in Figure 4d. Our proximity function predicts a qualitatively meaningful goal proximity: high estimated proximity near the goal and lower estimated proximity when the agent is farther away from the goal. The corners of the rooms show low goal proximity since less expert trajectories pass over those regions compared to the center of each room.

Finally, we investigate our hypothesis that the goal proximity function allows for greater generalization, which results in better task performance with smaller demonstration coverage. To test this hypothesis, we compare the cases where extreme (25% coverage), moderate, and no generalization (100% coverage) are required. Figure 4 demonstrates that our method consistently achieves 100% success rates in 3M steps with 50%-100% demonstration coverages and is not as affected by increasingly difficult generalization as baselines. In contrast, GAIL and all LfO baselines fail to learn the NAVIGATION task when expert demonstrations do not cover all configurations. This supports our hypothesis that the goal proximity function is able to capture the task structure and therefore, generalize better to unseen configurations.

## 4.4 ROBOT MANIPULATION

We further evaluate our method in two continuous control tasks: FETCH PICK and FETCH PUSH from Plappert et al. (2018). In the FETCH PICK task shown in Figure 2b, a Fetch robotic arm must

grasp and move a block to a target position. In FETCH PUSH, the Fetch arm pushes a block to a target position, as shown in Figure 2c. Both the initial position of the block and target are randomly initialized. For each, we provide 1k demonstrations, consisting of 33k and 28k transitions for FETCH PICK and FETCH PUSH respectively, generated using a hand-engineered policy . We create a 50% holdout set of starting states for agent learning by splitting the continuous state space into a 4 by 4 grid and holding out two cells per row to sample the block and target starting positions from.

In the FETCH PICK task, our method achieves more than 80% success rate while the success rates of GAIfO and GAIfO-s are upper-bounded by 50% due to the limited coverage of expert demonstrations (see Figure 3b). Our method learns slower than GAIL but achieves comparable final performance even though GAIL learns from expert actions. The FETCH PUSH task is more challenging than FETCH PICK due to the more complicated contact dynamics for pushing interactions. In Figure 3c, the demonstrations are collected with full coverage but the policy is trained in a version of the environment with 2 times larger noise applied to the starting state. All methods fail to learn diagonal pushing movements but our method still learns horizontal pushing faster and achieves higher performance than all other baselines. We evaluate both FETCH tasks under two different generalization setups, different demonstration coverages (Figure 8) and different amounts of noise (Figure 9), and the results consistently show that our proximity function is able to accelerate policy learning in continuous control environments with superior generalization capability.

## 4.5 ANT LOCOMOTION

We used the ANT REACH environment proposed in Ghosh et al. (2018), simulated in the MuJoCo physics engine (Todorov et al., 2012). In this task, the quadruped ant is tasked to reach a randomly generated goal, which is along the perimeter of a half circle of radius 5m centered around the ant (see Figure 2d). We provide 1k demonstrations, which contain 25k transitions in total. When demonstrations are collected, no noise is added to the initial pose of the ant whereas random noise is added during policy learning, which requires the reward functions to generalize to unseen states.

In Figure 3d, with 0.03 added noise, our method achieves 35% success rate while BCO, GAIfO, and GAIfO-s achieve 1%, 2%, and 7%, respectively. This result illustrates the importance of learning proximity over learning to discriminate expert and agent states for generalization to unseen states. The performance of GAIfO and GAIfO-s drops drastically with larger joint angle randomness as shown in Figure 9. As the ANT REACH task is not as sensitive to noise in actions compared to other tasks, BC and GAIL show superior results but our method still achieves comparable performance.

We also ablate the various aspects of our method in Figure 5. First, we verify the effect of the uncertainty penalty used in the proximity reward. The learning curves with different $\lambda$ are plotted in Figure 5a and demonstrate that our method works the best with $\lambda = 0.1$. Both too low and too high uncertainty penalties degrade the performance. Figure 5b shows the linearly discounted proximity function learns marginally faster than the exponentially discounted proximity function. In Figure 5c, we test the importance of online and offline training of the proximity function. The result shows that the agent fails to learn the task without online updates using agent trajectories. Meanwhile, no proximity function pre-training lowers performance.

## 4.6 ABLATION STUDY

Finally, we analyze the contribution of the proximity function, reward formulation, and uncertainty to our method's performance in Figure 6. Adding uncertainty to GAIfO-s (GAIfO-s+Uncert) produced a 5.8% boost in average success rate compared to regular GAIfO-s, which is not a significant improvement. Proximity supervision, without the uncertainty penalty, resulted in a 28.1% increase in average performance over GAIfO-s with the difference-based reward $R(s_t, s_{t+1}) = f(s_{t+1}) - f(s_t)$ (Prox+Diff) and 15.9% with the absolute reward $R(s_t) = f(s_t)$ (Prox+Abs). This higher performance means modeling proximity is more important than using the uncertainty penalty for our method.

We also found that the uncertainty penalty and proximity function have a synergistic interaction. Combining both the proximity and uncertainty gives a 43.3% increase with the difference-based reward (Prox+Diff+Uncert) and 33.0% increase with the absolute reward (Prox+Abs+Uncert). We can observe that the difference-based reward consistently outperforms the absolute reward except on ANT REACH, where the bias of the absolute reward Kostrikov et al. (2019) helps the agent survive

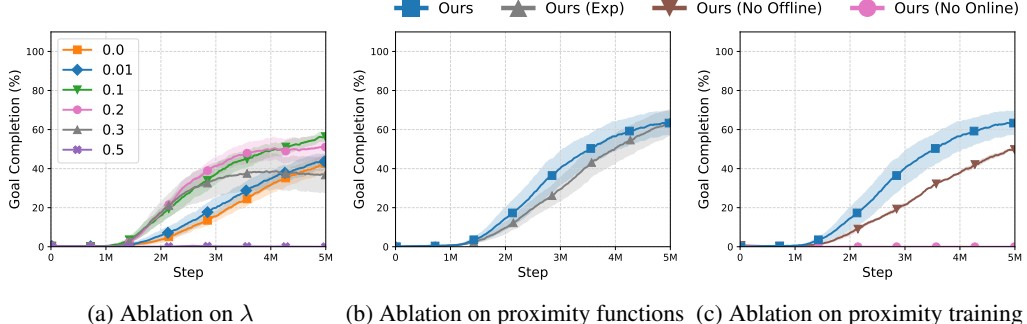

(a) Ablation on $\lambda$     (b) Ablation on proximity functions     (c) Ablation on proximity training

Figure 5: Ablation analysis of our method on ANT REACH. (a) Comparing different $\lambda$ values to show the effect of the uncertainty penalty. $\lambda = 0$ corresponds to no uncertainty penalty. (b) Contrasting two alternate formulations of the proximity function. (c) Analyzing the effect of online and offline proximity function training.

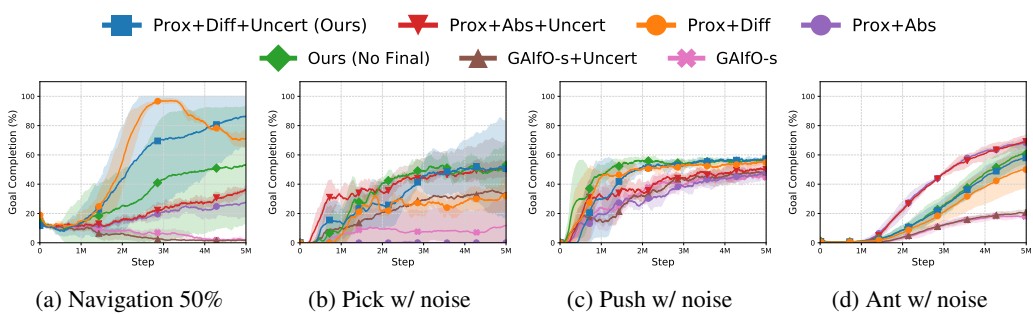

(a) Navigation 50%     (b) Pick w/ noise     (c) Push w/ noise     (d) Ant w/ noise

Figure 6: Ablation analysis of the contribution of proximity, uncertainty penalty, and reward formulation to our method's performance. "Prox" uses the goal proximity function while "GAIfO-s" does not. "+Diff" uses $R(s_t, s_{t+1}) = f(s_{t+1}) - f(s_t)$ and "+Abs" uses $R(s_t) = f(s_t)$ as the per-time step reward. "+Uncert" adds the uncertainty penalty to the reward. Finally, "No Final" removes the sparse proximity reward at the final time step.

longer and reach the goal. Firstly, this shows the uncertainty penalty is more important for the proximity function as it models fine-grain temporal information where inaccuracies can be exploited, as opposed to the binary classification given by other adversarial imitation learning discriminators. Secondly, both with and without the uncertainty penalty, the difference-based proximity reward performs better than the absolute proximity reward. In conclusion, all three components of proximity, uncertainty, and difference-based reward are crucial for our method.

In Figure 6, we also evaluate the advantage of the additional sparse proximity reward given at the final time step. Compared to our method without this final reward, it results in a minor 0.9% average performance improvement, meaning this component is not critical to our method.

## 5 CONCLUSION

In this work, we propose a learning from observation (LfO) method inspired by how humans acquire skills by watching others performing tasks. Specifically, we propose to learn a goal proximity function from demonstrations which provides an estimate of temporal distance to the goal. Then, we utilize this learned proximity to encourage the policy to progressively move to states with higher proximity and eventually reach the goal. The experimental results on navigation, robotic manipulation, and locomotion tasks demonstrate that our goal proximity function improves the generalization capability to unseen states, which results in better learning efficiency and superior performance of our method compared to the state-of-the-art LfO approaches. Moreover, our method achieves comparable performance with LfD approaches.

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

## A    Overview

As supplementary material for enhancing the main paper, we provide the following information:

- **Further analysis**: We first provide analyses of how our model and baselines generalize to unseen states. We also include qualitative results demonstrating what the proximity function learns. Finally, we show ablation experiments on additional environments.

- **Implementation details**: We describe the environments and provide training hyper-parameters and architectures for our models.

- **Code**: Complete code of our model, environments, and experiments can be found in the `code` directory. The `code/README.md` file documents the installation process, how to download the expert datasets, and commands for running experiments.

- **Video**: We provide the `results.mp4` file to present qualitative results of our method and baseline methods.

## B    Comparison with GAIL

Our method shares a similar adversarial training process with GAIL. The following steps describe how to achieve our method starting from GAIL. Firstly, like the discriminator in GAIfO-s, our proximity function takes only states as input. Next, rather than training the discriminator to classify expert from agent, we train the proximity function to regress to the proximity labels which are 0 for agent and the time discounted value between 0 and 1 for expert. Our reward formulation also differs from GAIL approaches which give a log probability reward based on the discriminator output. We instead incorporate a proximity estimation uncertainty penalty, a difference-based reward, and a sparse reward given for the proximity of the final state, as shown in Equation 2.

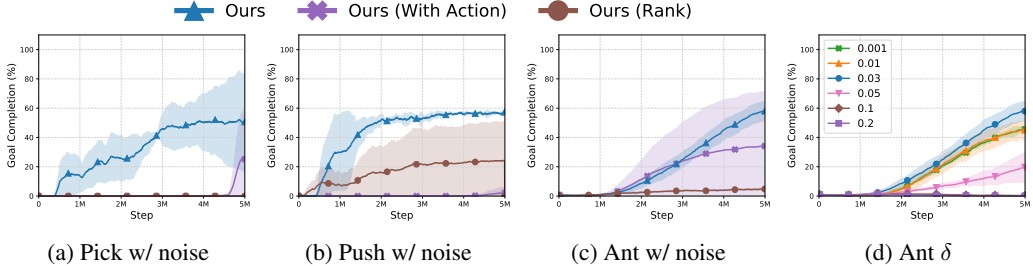

(a) Pick w/ noise          (b) Push w/ noise          (c) Ant w/ noise          (d) Ant $\delta$

Figure 7: Examining different ways of training the proximity function. (a-c) Comparing training the proximity function with actions as input or with a ranking-based objective. (d) Analyzing different choices of $\delta$ for training the proximity function. The model learns similarly well over a range of $\delta$ values around $1/H$, which in Ant Reach is 0.02, but struggles for large $\delta$ as many proximity values will be clipped to 0.

## C    Analysis on Generalization of Our Method and Baselines

By learning to predict the goal proximity, the proximity function not only learns to discriminate expert and agent states but also models task progress, which provides more information about the task. With additional supervision on learning goal proximity, we expect the proximity function to provide a more informative learning signal to the policy and generalize better to unseen states than baselines which overfit the reward function to expert demonstrations. To analyze how well our method and the baselines can generalize to unseen states, we vary the difference between the states encountered in expert demonstrations and agent learning.

One way, we vary the difference between expert demonstrations and agent learning is by restricting the expert demonstrations to only cover parts of the state space. For Navigation, Fetch Pick and Fetch Push we show results for demonstrations that cover 100% 75%, 50% and 25% of the state

space. For the discrete state space in NAVIGATION we restricted expert demonstrations to the fraction of possible agent start and goal configurations. For the two continuous state FETCH tasks, we break the 2D continuous sampling region a 4 by 4 grid and hold out one cell per row for 75% coverage and three cells per row for 25% coverage to sample the block and target starting positions from.

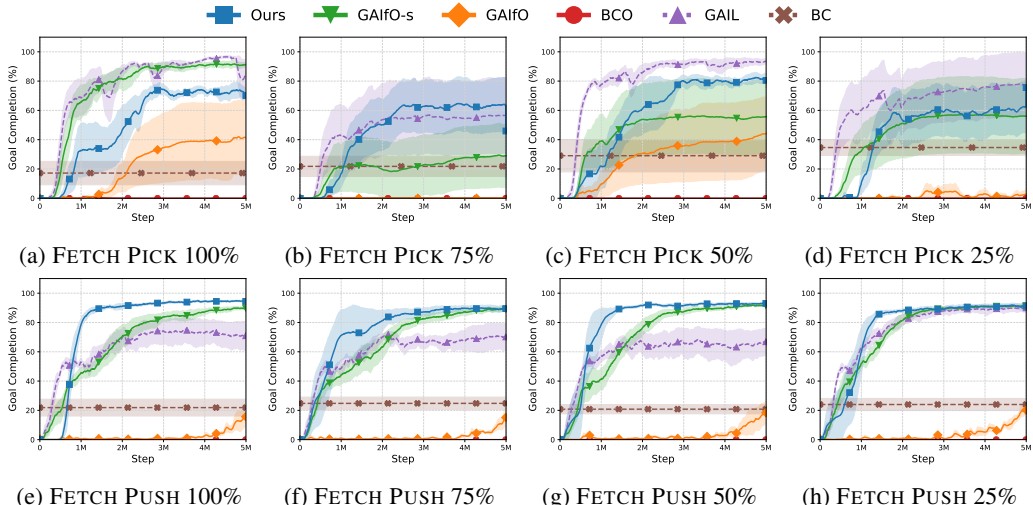

(a) FETCH PICK 100%    (b) FETCH PICK 75%    (c) FETCH PICK 50%    (d) FETCH PICK 25%

(e) FETCH PUSH 100%    (f) FETCH PUSH 75%    (g) FETCH PUSH 50%    (h) FETCH PUSH 25%

Figure 8: Analyzing generalization under different demonstration coverages. The percentage refers to the percentage of regions the expert demonstrations cover. A higher coverage percentage indicates less generalization, with 100% requiring no generalization.

Likewise, we also measured generalization by varying the difference between expert demonstrations and agent learning by increasing the initial state noise during agent learning. On FETCH PICK, FETCH PUSH and ANT REACH we show results for four different noise settings for the two FETCH tasks, the 2D sampling region is scaled by the noise factor. For ANT REACH, uniform noise, scaled by the noise factor, is added to the initial joint angles, whereas the demonstrations have no noise. If our method allows for greater generalization from the expert demonstrations, our method should perform well even under states different than those in the expert demonstrations.

The results of our method and baselines across varying degrees of generalization are shown in Figure 8 and Figure 9. Note that the results in the main paper are for 50% coverage in FETCH PICK, 2x noise in FETCH PUSH, and 0.05 noise in ANT REACH. Across both harder and easier generalization, our method demonstrates more consistent performance. While GAIfO-s performs well on high coverage, which requires little generalization in agent learning, its performance deteriorates as the expert demonstration coverage decreases.

## D   QUALITATIVE RESULTS

In this section, we aim to qualitatively verify if the learned goal proximity function gives a meaningful measure progress towards the goal for the agent. It is important for agent learning, that this proximity function gives higher values for states which are temporally closer to the goal. To verify these intuitions, we visualize the proximity values predicted by the proximity function in a successful episode from agent learning in Figure 10.

From the qualitative results of FETCH PICK and FETCH PUSH in Figure 10, we can observe that as the agent moves the block closer to the goal, the predicted proximity increases. This provides an example of the proximity function generalizing to agent experience and providing a meaningful reward signal for agent learning. We notice that while the predictions increase as the agent nears the goal, the proximity prediction values are often low ($<0.1$) as in Figure 10a. We hypothesize this is due to the adversarial training which labels agent experience with 0 proximity and lowers the average proximity predicted across states. For videos of more qualitative evaluations for our method and baselines, refer to `results.mp4`, also included in the submission.

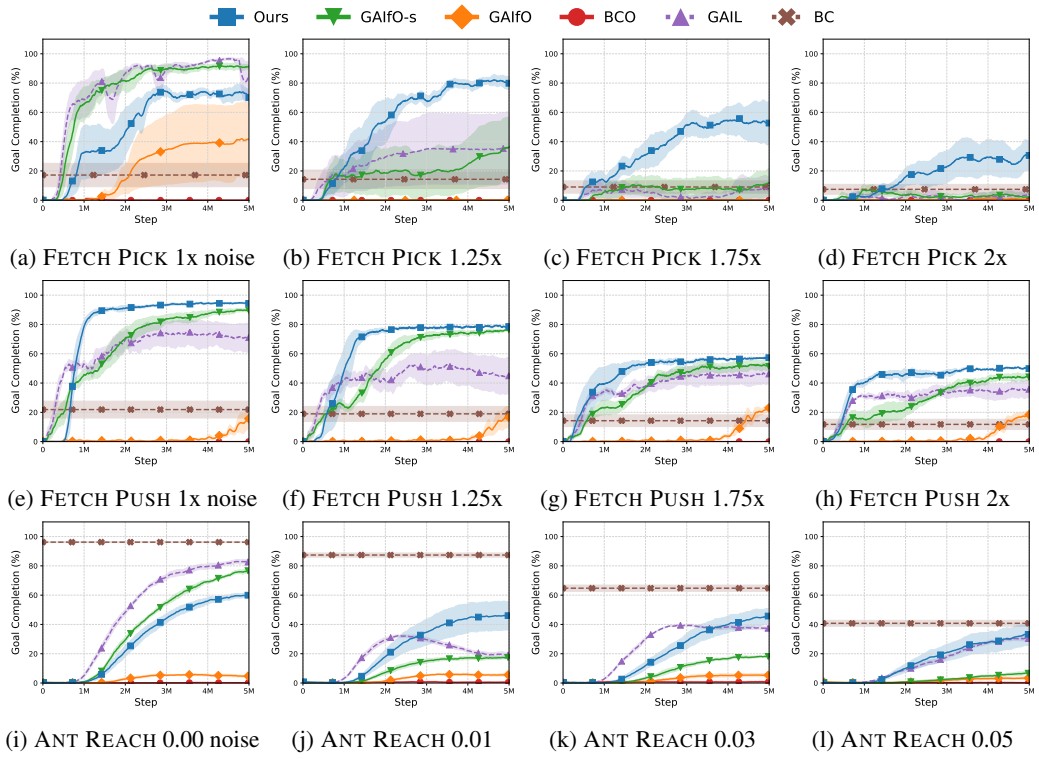

Figure 9: Analyzing generalization to more noisy environments. The number indicates the amount of additional noise in agent learning compared to that in the expert demonstrations, with more noise requiring harder generalization.

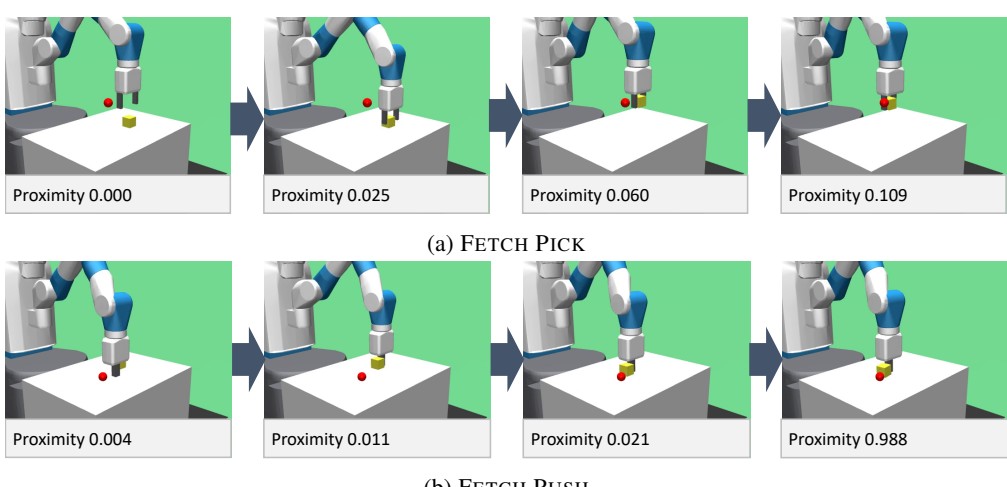

Figure 10: Visualizing the proximity predictions for a successful trajectory from agent learning in (a) FETCH PICK and (b) FETCH PUSH. Four informative frames are selected from the overall trajectory and the predicted proximity value is displayed below.

## E    FURTHER ABLATIONS

We include additional ablations to further highlight the advantages of our main proposed method over its variants. We evaluate against the same ablations proposed in the main paper, but across all environments. We present all these experiments in Figure 11.

We also attempted to compare to an ablation which learns the proximity function through a ranking based loss from Brown et al. (2019). However, we empirically found it to be ineffective and therefore did not include it. This ranking based loss uses the criterion that for two states from an expert trajectory $s_{t_1}, s_{t_2}$, the proximities should obey $f(s_{t_1}) < f(s_{t_2})$ if $t_1 < t_2$. We therefore train the proximity function with the cross entropy loss $-\sum_{t_i < t_j} \log \frac{\exp f_\phi(s_{t_j})}{\exp f_\phi(s_{t_i}) + \exp f_\phi(s_{t_j})}$. We incorporate agent experience by adding an additional loss which ranks expert states above agent states for randomly sampled pairs of expert and agent states $(s_e, s_a)$, through the cross entropy loss $-\sum_{s_a \sim D^e, s_e \sim \pi_\theta} \log \frac{\exp f_\phi(s_e)}{\exp f_\phi(s_a) + \exp f_\phi(s_e)}$.

Unlike the discounting factor in the discounting based proximity function, the ranking based training requires no hyperparameters. However, the lack of supervision on ground truth proximity scores could result in less meaningful predicted proximities and a worse learning signal for the agent, which could explain the observed poor performance.

As seen from the additional ablation analysis in Figure 11, our method has the best performance in the majority of environments. In each task, incorporating uncertainty and online updates is crucial. Our main method using the linear proximity function always performs slightly better than the exponential proximity function. Using offline updates is helpful in all environments except NAVIGATION.

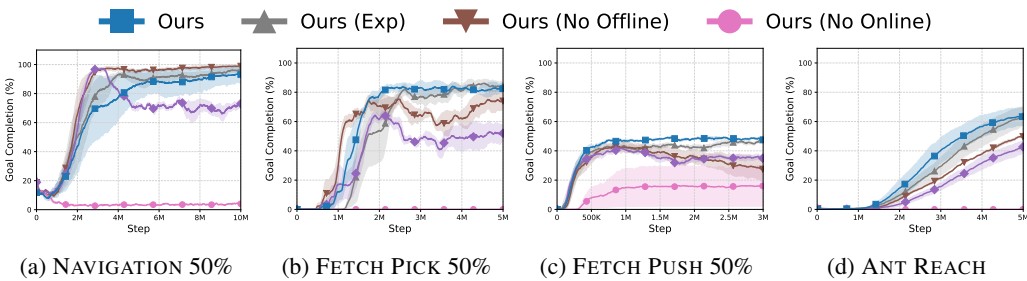

(a) NAVIGATION 50%    (b) FETCH PICK 50%    (c) FETCH PUSH 50%    (d) ANT REACH

Figure 11: Ablation experiments across additional environments. Our method shows consistently superior performance over all ablations.

## F    FURTHER BASELINES

In Figure 12, we compare to SQIL (Reddy et al., 2020), an imitation learning approach which demonstrates higher sample efficiency with off-policy RL. SQIL modifies the replay buffer with the expert demonstrations which are assigned reward $+1$ while all agent experience is assigned reward 0. We use Soft Actor-Critic (Haarnoja et al., 2018) using the same hyperparameters as in Reddy et al. (2020).

Our method consistently outperforms SQIL in the Fetch tasks, despite SQIL using actions whereas our method does not. This can be because the Fetch tasks require precise actions (i.e., it is difficult to reverse moving the block in the wrong direction) which results in covariate shift that negatively impacts BC and SQIL (a form of regularized BC). On the other hand, SQIL learns the Ant Reach task well and performs similar to the behavioral cloning (BC) baseline since Ant Reach is not sensitive to small errors (i.e., the agent can recover from bad actions and change heading). We observed unstable SQIL results in Figure 12c, which can also be observed in Figure 2,3 of (Reddy et al., 2020), further training or hyperparameter tuning could stabilize training.

## G    IMPLEMENTATION DETAILS

### G.1    ENVIRONMENT DETAILS

The implementations of NAVIGATION, FETCH (which includes FETCH PICK and FETCH PUSH), and ANT REACH environments are based on Chevalier-Boisvert et al. (2018), Plappert et al. (2018), and Ghosh et al. (2018), respectively. The actions in the FETCH experiments use end effector position control and continuous control for the gripper. A 15-dimensional state in FETCH tasks consists of

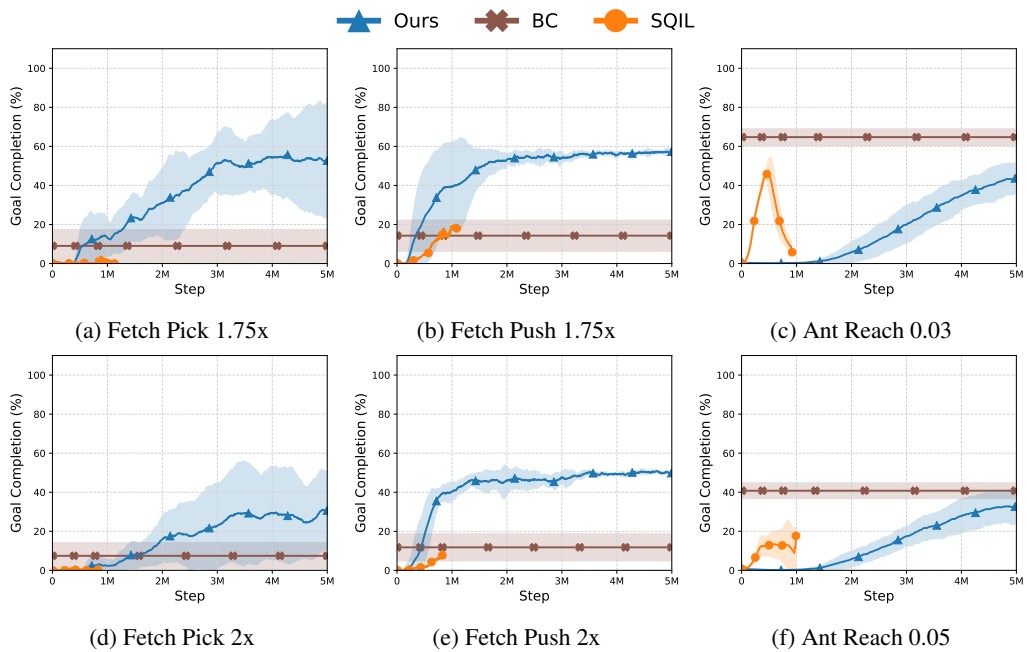

Figure 12: Comparing our method to SQIL across three tasks in the most challenging generalization settings. SQIL results are for 3 seeds while our method and BC are for 5 seeds. Note that SQIL and BC use actions whereas our method does not.

the relative position of the goal from the object, relative position of the end effector to the object, and robot state. We found that not including the velocity information was beneficial for all learning from observation approaches. Meanwhile, in NAVIGATION the state consists of a one-hot vector for each grid cell encoding wall, empty space, agent or goal. Finally, in ANT REACH the state consists of velocity, force and the relative goal position with the action space consisting of joint control. Each environment also randomly initializes the starting state and goal. The details of observation spaces, action spaces, and episode lengths are described in Table 1. All units in this section are in meters unless otherwise specified.

For NAVIGATION we collect 250 expert demonstrations and for all other environments we collect 1000 expert demonstrations. In NAVIGATION we use BFS search to collect expert demonstrations. In FETCH PICK we generate demonstrations by hard coding the Sawyer Robot to first reach above the object, then reach down and grasp, and finally move to the target position. Similarly, in FETCH PUSH, we reach behind the object and then execute a planar push forward towards to the goal. For ANT REACH, we collect demonstrations using an expert policy trained using PPO (Schulman et al., 2017) based on the reward function $R(s, a) = 1 - 0.2 \cdot ||p_{ant} - p_{goal}||_2 - 0.005 \cdot ||a||_2^2$, where $p_{ant}$ and $p_{goal}$ are $(x, y)$-positions of the ant and goal, respectively, and $a$ is an action. Please refer to the code for more details.

To evaluate the generalization capability of our method and baselines, we constrain the coverage of expert demonstrations or add additional starting state noise during agent learning as discussed in Section C.

Table 1: Environment details. In NAVIGATION the goal and agent are randomly initialized anywhere on the grid. The goal and object noise in FETCH describes the amount of uniform noise applied to the $(x, y)$ coordinates. In ANT REACH, the angle of the goal and the velocity of the agent are randomly initialized.

|  | NAVIGATION | FETCH PICK | FETCH PUSH | ANT REACH |
|---|---|---|---|---|
| Observation Space | (19,19,4) | 16 | 16 | 132 |
| Goal Noise | - | $(x,y) \in [\pm 0.02, \pm 0.05]$ | $(x,y) \in [\pm 0.02, \pm 0.05]$ | $\theta \in [0, \pi]$ |
| Object / Agent Noise | - | $(x,y) \in [\pm 0.02, \pm 0.02]$ | $(x,y) \in [\pm 0.02, \pm 0.05]$ | $v \in [\pm 0.005]$ |
| Action Space | 4 | 4 | 3 | 8 |
| Episode length | 50 | 50 | 60 | 50 |

## G.2 NETWORK ARCHITECTURES

**Actor and critic networks:** We use the same architecture for actor and critic networks except for the output layer where the actor network outputs an action distribution while the critic network outputs a critic value. For NAVIGATION, the actor and critic network consists of $CONV(3, 2, 16) - ReLU - MaxPool(2, 2) - CONV(3, 2, 32) - ReLU - CONV(3, 2, 64)$ followed by two fully-connected layers with hidden layer size 64, where $CONV(k, s, c)$ represents a $c$-channel convolutional layer with kernel size $k$ and stride $s$. For other tasks, we model the actor and critic networks as a two separate 3-layer MLP with hidden layer size 256. For the continuous control tasks, the layer of the actor MLP is two-headed to output the mean and standard deviation of the action distribution.

**Goal proximity function and discriminator:** The goal proximity function and discriminator use a CNN encoder followed by a hidden layer of size 64 for NAVIGATION and a 3-layer MLP with a hidden layer size of size 64 for other tasks.

## G.3 TRAINING DETAILS

For our method and all baselines except BC (Pomerleau, 1989) and BCO (Torabi et al., 2018a), we train policies using PPO. The hyperparameters for policy training are shown in Table 2, while the hyperparameters for the proximity and discriminator function are shown in Table 3. For our method, in some tasks, we also found it slightly helpful to scale the reward by a constant factor and these values are also included in Table 2.

In BC, the demonstrations were split into 80% training data and 20% validation data. The policy was trained on the training data until the validation loss stopped decreasing. The policy is then evaluated for 1,000 episodes to get an average success rate. In GAIfO-s and GAIL the AIRL reward from Fu et al. (2018) is used.

Table 2: PPO hyperparameters.

| Hyperparameter | Value |
|---|---|
| Learning Rate | 0.001 (NAVIGATION, FETCH PUSH), 0.0003 (ANT, FETCH PICK) |
| Learning Rate Decay | linear decay |
| # Mini-batches | 32 (FETCH, ANT), 4 (NAVIGATION) |
| # Epochs per Update | 10 (FETCH, ANT), 4 (NAVIGATION) |
| Discount Factor | 0.99 |
| Rollout Size | 4,096 (NAVIGATION, FETCH), 16,000 (ANT) |
| Entropy Coefficient | 0.001 (ANT), 0.01 (NAVIGATION, FETCH) |
| Reward Scale | 1 (NAVIGATION), 10 (FETCH), 50 (ANT REACH) |
| State Normalization | True (FETCH, ANT), False (NAVIGATION) |

Table 3: Hyperparameters for goal proximity functions and discriminators.

| Hyperparameter | Value |
|---|---|
| # Networks for Ensemble | 5 |
| Uncertainty Coefficient $\lambda$ | 0.01 (ANT, FETCH PUSH), 0.1 (FETCH PICK, NAVIGATION) |
| Discounting Factor $\delta$ | 0.01 (NAVIGATION, ANT REACH), 0.02 (FETCH PICK, FETCH PUSH) |
| Learning Rate (Ours) | 0.001 (NAVIGATION, FETCH), 0.0001 (ANT) |
| Learning Rate (GAIfO-s) | 0.0001 |
| Learning Rate (GAIfO) | 0.001 |
| Batch Size | 128 (FETCH, ANT), 32 (NAVIGATION) |
| # Updates per Agent Update | 1 |
| Agent Experience Buffer Size | 16,000 (ANT), 4096 (NAVIGATION, FETCH) |

