# OpenReview forum: "Goal-Driven Imitation Learning from Observation by Inferring Goal Proximity"
_ICLR.cc/2021/Conference — Reject_

### Official Review · AnonReviewer4 · 2020-10-26

**Rating:** 6
**Confidence:** 3

**Review:**

**Paper summary**

This paper proposes a method for imitation learning from observations based on learning a goal proximity function from expert demonstrations and using it as a dense reward for training an imitator. The authors show that this method improves generalization to unseen states in comparison to several baselines.

**Pros:**

- The idea is simple, well-motivated and the paper is easy to read, clear and well-written.

- The experiments are well designed and the results are explained with sufficient detail.

**Cons:**

While I find the idea interesting and the experiments well-designed, I have a few concerns about the method and would need some clarifications to evaluate it further:

1. Figure 5 and 9 demonstrate that a key component of the method is the adversarial training of the proximity function. Without it, the method completely fails. In comparison, not using the uncertainty part, or not training offline, makes less of a difference. This seems to indicate that the adversarial part is more important than the temporal aspect of it. If we ignore the temporal aspect and replace the target for f, $1-\delta (T-t)$, by its upper value of 1, we obtain a loss: $E_{expert} [f -1]^2 + E_{imitator} f^2$ which is very similar to the one used in GAIL: $E_{expert} \log(\frac{1}{1-D}) + E_{imitator} \log \frac{1}{D})$ for values of $D$ and $f$ between 0 and 1 (plotting the 4 functions recommended). Given this similarity, I think the paper would benefit from one more ablation of this form: removing the temporal part and keeping the adversarial component.

2. Following from point 1, I wonder why are the GAIL results so different (e.g GAIfO). One concern I have is the final reward. The method proposed in this paper makes the assumption that the last state of the demonstration is the goal. Therefore the authors use an extra reward at this time step (Eq. 2). If I understand correctly, GAIL in general doesn’t make this assumption, so I wonder if this is at least in part responsible for the differences between these methods. I think the paper would benefit from clarifying this by performing an ablation that removes this final reward.

3. One maybe more superficial concern is whether the name ‘goal proximity’ is appropriate. Given that the function $f$ is not only trained to follow time, but also adversarially to be zero on imitator data, this name is confusing to me. In fact, as shown in Fig 4d, the value of $f$ doesn’t end up corresponding to the time (or number of actions) that would be required to reach the goal in the general sense.

4. Regarding Fig 4d. I’m not sure why the centers of all quadrants are light. Shouldn’t only the path to the goal used by the expert be light?

---

> ### Author Response · Authors · 2020-11-19
> **Response to Reviewer 4**
>
> We thank the reviewer for the constructive feedback and address the concerns in detail below.
>
> **Perform an ablation that removes the temporal part and keeps the adversarial component.**
> We thank Reviewer 4 (R4) for suggesting an important ablation study. Our method without the temporal discounting and difference-based reward is the same as GAIfO-s with uncertainty penalty. We compare our method to this approach in an additional experiment in Figure 6 which evaluates the value of temporal discounting. These results illustrate that temporal discounting, the core contribution of our method, indeed drives performance and outperforms the one without the temporal part by 37.5% average success rate across all tasks.
>
>
> **Perform an ablation that removes the final reward.**
> We agree that this is again an important ablation. Thus, we performed an ablation removing the final reward and added the result in Figure 6. Using the final reward only marginally increases performance by 0.9%, which tells us this additional reward is not a critical component for the improved performance of our method.
>
>
> **“One maybe more superficial concern is whether the name ‘goal proximity’ is appropriate.”**
> We agree that $f$ does not exactly capture the true goal proximity due to the adversarial training. However, using goal proximity is a key intuition of our method, and contrary to prior methods $f$ is trained with goal proximity supervision. This manifests in the interpretation of Figure 4d where the areas more commonly encountered by the expert have higher proximity.
>
>
> **Why are all quadrant centers light in Fig 4d?**
> The expert demonstrations often pass through the quadrant centers when navigating between rooms. This provides more labeled proximity values greater than 0 for the quadrant centers. At the same time, the agent can easily get stuck near the walls during exploration and thus provides more samples with proximity value 0 around the walls. Due to this biased data, Figure 4d shows higher proximity values in centers of all quadrants.

---

### Official Review · AnonReviewer2 · 2020-10-27
**Nice work, but missing crucial literature proposing temporal progress rewards**

**Rating:** 6
**Confidence:** 5

**Review:**

## Paper and Review Summary

This paper introduces a goal proximity approach to learning rewards from observations captured during demonstrations in goal-oriented tasks. This metric allocates rewards based on the temporal distance to a goal, relying on a model trained to predict this distance. The paper also proposes an adversarial learning approach to policy optimisation with this reward, which helps to improve policies in regions where demonstrations were not captured.

I like the idea, but, unfortunately, this paper has missed key related work [Angelov et al.], [Burke et al.] on reward inference and policy scheduling using temporal progress metrics, which have been proposed previously. Although I do see potential novelty and additions going beyond the work in [Angelov et al.], [Burke et al.] (adversarial training, ablations the incorporation of uncertainty in the metric) - the primary contribution is severely reduced. As a result, I am inclined to recommend *rejecting* this work, unless the paper revisions can convince me that there are sufficient differences and novelty. However, this may require a substantial rewrite of the introduction and conclusions and major revisions.

## Pros

- The paper is well written, with detailed experiments and nice ablations
- I like the idea of combining a goal progress metric with adversarial learning
- The inclusion of the uncertainty metric is interesting, potentially allowing for risk-based policy optimisation

### Cons
Limited novelty:
- A linear goal progress metric has already proposed by Angelov et al. in the paper [Composing Diverse Policies for Temporally Extended Tasks](https://arxiv.org/pdf/1907.08199.pdf). This paper trains a model to predict the normalised time to goal using observed demonstrations, and uses this to select sub-policies for long-horizon tasks, in a model-based setting. Angelov et al. do not optimise policies directly using the goal metric.
- In follow on work by Burke et al. in the paper [Learning rewards for robotic ultrasound scanning
using probabilistic temporal ranking](https://arxiv.org/abs/2002.01240) this goal progress metric of Angelov et al. is extended to a probabilistic temporal ranking metric that allows for non-monotonically increasing goal progress in demonstrations to be captured. Here, the goal progress reward is used for policy optimisation (value iteration for grid world experiments) and online learning.

### Questions

- On page 3, the paper states "*There are alternative ways to represent and learn goal proximity, such as exponentially discounted proximity and ranking-based proximity (Brown et al., 2019). But, in our experiments, linearly discounted proximity consistently performed better than alternatives*" - I am curious that linear proximity models performed better as experiments in Brown et al. and Burke et al. show that non-linear temporal progress metrics are more effective. In particular, ranking-based methods allow for non-monotonically increasing progress metrics. Is this comment based only on the comparison between exponentially and linearly increasing progress, or was a temporal ranking approach considered? If not, I would recommend exploring a temporal ranking approach going forward.
- In Equation (2), I see the progress metric is not used directly, instead a derived reward was used. Did you experiment with using the progress metric directly instead?
- Page 7 FETCH-PUSH - Why can't the methods learn diagonal pushing policies?
- Figure 5 - Why does the policy using offline only training fail completely?  Is this a function of the way/ amount the training data was collected or something else?

---

> ### Author Response · Authors · 2020-11-19
> **Response to Reviewer 2**
>
> We thank the reviewer for the constructive feedback and address the concerns in detail below.
>
> **Limited novelty with missing relevant work [1, 2].**
> We thank Reviewer 2 (R2) for pointing out the highly relevant papers [1, 2], which highlight the value of modeling goal proximity (task progress) from demonstrations. We would like to note that our primary contribution is not just about learning goal proximity, but about proper proximity reward design, which measures an improvement in goal proximity, together with the adversarial imitation learning scheme and uncertainty penalty. With exhaustive evaluation, we show that all these components are essential to achieve better performance and generalizability of imitation learning.
>
> Angelov et al. [1] utilizes the learned proximity to choose a proper sub-policy but does not train a policy from the learned proximity. On the other hand, Burke et al. [2] proposes to learn a reward function using a ranking model and use it for policy optimization, similar to our method, and demonstrates the advantage of using goal proximity as a reward for training a policy. However, the reward signal used in [2] is trained only from the expert demonstrations and can fail to generalize to agent experience, allowing the agent to exploit inaccurate proximity predictions for reward. In contrast, our method proposes to (1) learn to make progress by using difference-based proximity reward, (2) directly supervise the goal proximity with the temporal distance to the goal, (3) jointly train the proximity function and policy, and (4) incorporate an uncertainty penalty.
>
> We cited both papers [1, 2] in the related work section along with the discussion of other prior work which also learned goal proximity. We will also soon update the abstract, introduction, and conclusion to differentiate our contribution relative to [1,2].
>
>
> **Was a temporal ranking-based proximity empirically compared?**
> The comment about the ranking-based proximity was made based on our empirical evaluation. As R2 mentioned, the temporal ranking-based methods [2, 3] can learn non-linear and non-monotonically increasing progress metrics. However, we empirically found that the ranking-based proximity [3] does not perform as well as linearly and exponentially discounted proximity. For optimal demonstrations the proximity function with direct supervision on the proximity score can be easier to learn than relative rankings. We added the comparison with the ranking-based proximity in Figure 7a-c. The probabilistic ranking model of [2] is a plausible future direction to explore for ranking-based proximity.
>
>
> **“In Equation (2), I see the progress metric is not used directly, instead a derived reward was used. Did you experiment with using the progress metric directly instead?”**
> Thank you for pointing out the importance of our reward formulation, which is one of primary contributions of the paper. According to R2, we ran additional experiments without the proximity difference term in the reward and directly using the proximity (i.e., $R(s_t) = f(s_t)$). The results are added in Figure 6. Directly using proximity instead of the difference-based reward decreases success rates on average by 10.6% across all 4 tasks. This is because the difference-based reward at every time step encourages the agent to make consistent progress towards the goal. We also experimented without additional absolute proximity reward at the final time step (i.e., $R(s_t, s_{t+1}) = f(s_{t+1}) - f(s_t)$) and found this marginally hurts performance by 0.9%.
>
>
> **“FETCH-PUSH - Why can't the methods learn diagonal pushing policies?”**
> We found that diagonal pushes require longer and more complicated motions than pushing horizontally not because of learning algorithms but because of the rectangular shape of the object.
>
>
> **“Why does the policy using offline only training fail completely?”**
> The policy fails when the proximity function is trained only with the offline data since the proximity function does not generalize to sub-optimal agent experience and thus cannot guide the agent effectively. One of our core contributions is to continue training the proximity function with online trajectories using an adversarial objective.
>
>
> [1] Angelov et al., "Composing Diverse Policies for Temporally Extended Tasks", RA-L (2020)
>
> [2] Burke et al., "Learning robotic ultrasound scanning using probabilistic temporal ranking", arXiv preprint arXiv:2002.01240 (2020)
>
> [3] Brown et al., “Extrapolating beyond suboptimal demonstrations via inverse reinforcement learning from observations”, ICML (2019)

---

> > ### Comment · AnonReviewer2 · 2020-11-19
> > **Thank you for your detailed response and revisions**
> >
> > Thank you for your detailed response and engagement with the review process. See below for some comments and remaining concerns and queries.
> >
> > I believe that the paper has been greatly strengthened by the additional ablation experiments conducted around the contribution of the uncertainty, proximity and derived proximity components of the reward - these results are interesting.
> > - How exactly were the percentage improvements in 4.6 calculated? The ablation curves seem to indicate quite a bit of task variability in terms of which reward performs best - lending credence to other reviewer concerns about the heuristic nature of these rewards.
> > - Is it fair to say that this variability fundamentally arises from what is essentially task specific reward shaping? Ie. fixed linear/ exponentially proximity reward functions will always be task specific and the ordering of approaches will vary greatly based on the suite of tasks included in the benchmark, whereas a temporal ranking metric would be capable of adapting to the full range of desirable proximity rewards that may be required for tasks.
> > - Note: I don't necessarily feel that 'heuristic' is a bad thing, as there is value in empirical analysis of adversarial training and derived proximity rewards under what are a common class of problems, but it does mean that care needs to be taken about more general claims.
> >
> > Thank you for including the ranking reward in the GAIL ablations in Appendix B.
> > - how exactly is the rank computed here? Is this a temporal ranking metric, or a rank based on whether a trajectory was generated by an expert or the policy?
> > - Appendix E - Possibly clarify. As far as I understand, Brown et al. (2019), propose ranking across trajectories (not time steps), and Burke et al. (2020) apply this ranking idea to time steps - f(t1) > f(t2) if t1 < t2. However, as the authors point out in the rebuttal, this is only applied to expert demonstrations, and not using data gathered by the policy.

---

> > > ### Author Response · Authors · 2020-11-20
> > > **Follow Up Response to Reviewer 2**
> > >
> > > We thank the reviewer for the prompt response and follow up questions.
> > >
> > > **How were the percentage improvements in 4.6 calculated?**
> > > In Section 4.6, we took the difference between the goal completion percentage of GAIfO-s and the compared approach, and then took the average across all 4 tasks.
> > >
> > >
> > > **The ablation curves seem to indicate quite a bit of task variability in terms of which reward performs best.**
> > > Figure 6 shows that the performances of different reward formulations can be affected by the characteristics of the task. For example, in Ant Reach, the absolute proximity reward performs better than the difference-based reward since the absolute reward is always positive and implicitly biases the agent to survive longer and reach the goal [1].
> > >
> > > However, we would like to note that even with the task variability, we can see the tendency of the performance gain with goal proximity and uncertainty penalty. Moreover, our proposed difference-based reward works best in 3 of the 4 tasks.
> > >
> > >
> > > **A temporal ranking metric would be capable of adapting to the full range of desirable proximity rewards that may be required for tasks.**
> > > As R2 mentioned, the temporal ranking-based method [2, 3] could be more flexible to different tasks. However, it is not trivial to train the ranking-based proximity function with agent experience, which we show is crucial in Figures 5 and 11. This is because the rankings of the agent trajectories and the states within agent trajectories are unknown. We think this can be an interesting future work.
> > >
> > >
> > > **How is the rank computed?**
> > > We used a temporal ranking metric, which ranks the temporal ordering of states in a demonstration trajectory through a cross entropy loss similar to [3], except ranking states instead of trajectories as follows: $-\sum_{t_i < t_j} \log \frac{\text{exp} f_{\phi}(s_{t_j})}{\text{exp} f_{\phi}(s_{t_i}) + \text{exp} f_{\phi}(s_{t_j})}$.
> > >
> > > We incorporate agent experience by adding an additional loss which ranks expert states above agent states for randomly sampled pairs of expert and agent states $(s_e, s_a)$, through the cross entropy loss $-\sum_{s_a \sim D^e, s_e \sim \pi_{\theta}} \log \frac{\text{exp} f_{\phi}(s_e)}{\text{exp} f_{\phi}(s_a) + \text{exp} f_{\phi}(s_e)}$. We included this detail in the paper.
> > >
> > > [1] Kostrikov, et al., “Addressing sample inefficiency and reward bias in adversarial imitation learning”, ICLR (2019)
> > >
> > > [2] Burke et al., "Learning robotic ultrasound scanning using probabilistic temporal ranking", arXiv preprint arXiv:2002.01240 (2020)
> > >
> > > [3] Brown et al., “Extrapolating beyond suboptimal demonstrations via inverse reinforcement learning from observations”, ICML (2019)

---

> > > > ### Comment · AnonReviewer2 · 2020-11-20
> > > > **Thanks for the additional clarity**
> > > >
> > > > Thank you for the additional clarity and substantial improvements to the paper. I have changed my original review score to reflect this added work.

---

### Official Review · AnonReviewer1 · 2020-10-28
**Promising approach, but missing fundamental results**

**Rating:** 7
**Confidence:** 4

**Review:**

SUMMARY:
The authors propose a new method for imitation learning from observation that attempts to estimate and leverage a notion of goal proximity in order to help the learning process. The authors provide a framework for computing this estimate, and a technique for using that estimate -- along with a measure of uncertainty -- to perform imitation learning from observation. Experimental results for several domains are presented in which the proposed technique achieves better performance than the comparison methods.


STRENGTHS:
	(S1) The paper seeks to solve an interesting and relevant problem in IfO.
	(S2) The proposed technique -- estimating and using a notion of task completion proximity -- is (as far as I'm aware) a novel take on the IfO that seems to have the potential to advance the state of the art.


WEAKNESSES:
	(W1) Fundamental experimental results are missing. The paper proposes a technique with two major components: (a) an estimated proximity function, and (b) a method to exploit uncertainty information in that estimate. However, its not clear from the results which if it is the unique combination of these components that leads to the good results, or just one of them -- especially given the results in Figure 5 which shows how critical (b) is. One way to get at that would be to apply (b) to some adversarial imitation learning techniques (eg, GAIfO) and see how they perform. Without something like this, one cannot tell if it is the proximity function, the use of uncertainty information, or the combination that truly leads to improvement. This is a fundamental question that must be addressed.
	(W2) As written, some of the details of the proposed method are not clear to where it would likely be difficult to reproduce. For example:
		(a) There appears to be an unstated assumption that all demonstration trajectories terminate at a known time $T$, which is not typically true. Meanwhile, it seems like such trajectories would necessitate different choices of $\delta$, but the authors only discuss how to set $\delta$ in general according to some parameter $H$ which
		(b) It seems as though the training objective for $f_\phi$ articulated in Equation (1) is highly dependent on $\delta$, but it doesn't seem as though the choice of delta is discussed.


RECOMMENDATION STATEMENT:
The proposed method describes a novel approach to a good problem and seems to hold promise. However, I feel that important experimental results need to be added before the paper should be published.


QUESTIONS FOR AUTHORS:
	(Q1) How would a standard imitation from observation algorithm perform if endowed with the same uncertainty information as the proposed method?
	(Q2) In Figure 4d, why are cells in the NW quadrant seemingly just as proximal as cells in the SW/NE? It seems as though they should be less proximal.


MINOR COMMENTS:
	(MC1) The two legends present on Figure 5 are confusing while looking at (a), and its unclear where some of the curves in the legend at the top are in (b) and (c). The figure should be revised to be more clear.

---

> ### Author Response · Authors · 2020-11-19
> **Response to Reviewer 1**
>
> We thank the reviewer for the constructive feedback and address the concerns in detail below.
>
> **“One cannot tell if it is the proximity function, the use of uncertainty information, or the combination that truly leads to improvement.”**
> We agree that it is crucial to verify the importance of the goal proximity, uncertainty measure, difference-based reward, and their combination. As Reviewer 1 (R1) suggested, we did additional experiments with the best imitation from observation method (GAIfO-s) with uncertainty measure and presented the results in Figure 6.
>
> Across all tasks, adding uncertainty to GAIfO-s produced a 5.8% boost in average success rate compared to regular GAIfO-s, which is not a significant improvement. Proximity supervision, without the uncertainty penalty, resulted in a 28.1% increase in average performance over GAIfO-s with the difference-based reward $R(s, s’) = f(s’) - f(s)$ and 15.9% with the absolute reward $R(s) = f(s)$. This higher performance means modeling proximity is more important than using the uncertainty penalty for our method. Finally, combining both proximity and uncertainty gives a 43.3% increase with the difference-based reward and 33.0% increase with the absolute reward. Firstly, this shows uncertainty is more important for the proximity function as it models fine-grain temporal information where inaccuracies can be exploited, as opposed to the binary classification given by other adversarial imitation learning discriminators. Secondly, both with and without uncertainty, the difference-based reward performed better. In conclusion, all three components of: proximity, uncertainty and difference-based reward are crucial for our method.
>
>
> **Assumption that all demonstration trajectories terminate at a known time $T$.**
> We assume the episode terminates after achieving the goal, which results in a variable-length episode horizon $T$ in the expert demonstrations depending on how long the expert took to achieve the goal. On the other hand, $H$ is the maximum episode horizon, which is constant for each task. Hence, we used the maximum episode length $H$ to decide $\delta=1/H$, which fits the goal proximity into $[0, 1]$ for every state.
>
>
> **The choice of $\delta$ is not discussed.**
> We choose a single fixed value $\delta$ to scale the minimum possible proximity of states to 0, and thus use a $\delta$ of around $1/H$ for all experiments. In Figure 7d, we show that the model robustly learns for many $\delta$ values around $1/H$.
>
>
> **“In Figure 4d, why are cells in the NW quadrant seemingly just as proximal as cells in the SW/NE?”**
> During adversarial training, agent experience is labeled with proximity 0 which decreases the proximity estimates on the agent state distribution. In Figure 4d, it is possible the policy is biased to visit the SW and SE quadrants more which reduces the proximity. However, the proximity of the NW quadrant will be lowered after further training as the agent visits the quadrant more.
>
>
> **Legends in Figure 5 are confusing.**
> Thank you for the suggestion. We have updated the legend for Figure 5.

---

> > ### Comment · AnonReviewer1 · 2020-11-24
> > **Thanks**
> >
> > Thanks to the authors for the detailed response.
> >
> > The new results in Figure 6 are indeed compelling and help make the case for the unique combination of methods proposed in the paper.
> >
> > If the authors allow for variable-length episode horizons, then why is a single $T$ used in the first term in Equation (4)? Shouldn't the $T$ there be dependent upon which particular expert demonstration was drawn?
> >
> > Regarding $\delta$, I'd recommend adding a sentence to the manuscript somewhere to explicitly explain how $T$s and $H$ are related.

---

> > > ### Author Response · Authors · 2020-11-24
> > > **Follow Up Response to Reviewer 1**
> > >
> > > We thank the reviewer for the follow up questions and suggestions.
> > >
> > > **Why is a single $T$ used in the first term in Equation (4)?**
> > > R1 is correct that $T$ is a variable that depends on a particular expert demonstration. We added a sentence that explicitly refers to $T$ as variable depending on the expert demonstration.
> > >
> > >
> > > **Regarding $\delta$, I'd recommend adding a sentence to the manuscript somewhere to explicitly explain how $T$s and $H$ are related.**
> > > Thank you for the suggestion. We updated the paper accordingly.

---

> > > > ### Comment · AnonReviewer1 · 2020-11-25
> > > > **Thanks**
> > > >
> > > > Thanks for acknowledging this minor point, but looking at the latest manuscript, I don't believe it has yet been resolved.
> > > >
> > > > My own confusion stemmed from the fact that $T$ isn't subscripted with $i$ (i.e., $T_i$) in the relevant equations (e.g., (1) and (4)) and various points throughout the text. Would it not be appropriate to do so? Otherwise, it still appears as though $T$ is fixed across all demonstrations even though the authors here have acknowledged it may be different.

---

> > > > > ### Author Response · Authors · 2020-11-25
> > > > > **Follow Up Response to R1**
> > > > >
> > > > > We thank R1 for the suggestion and agree that changing $T$ to $T_i$ will make the paper more clear. We updated the paper accordingly.

---

### Official Review · AnonReviewer3 · 2020-10-29
**Theoretical foundation is better to be clarified.**

**Rating:** 5
**Confidence:** 3

**Review:**

To accelerate and improve imitation learning for goal-driven tasks, the authors introduce a goal proximity function that is learned from the observation of expert demonstrations and online agent experience.
The inferred goal proximity is used as an additional reward signal. The authors showed that heuristics efficiently improve the performance of imitation learning.

The method is simple and looks effective, as shown in the experiment.
However, from the theoretical viewpoint, this proposal looks a heuristic method.
It is better to clarify the theoretical foundation.

The relationship with GAIL is mentioned several times. However, the explicit comparison between the proposed method and GAIL is not given.
(For example, "First, we set the target proximity of states in agent trajectories to 0, similar to adversarial imitation learning methods (Ho & Ermon, 2016), and train the proximity function with both expert demonstrations and agent experience by minimizing the following loss." )
Describing the comparison (as an appendix) may help readers to understand the key idea.

To my understanding, the paper focus on LfO. However, the relationship between LfO and "goal proximity" is not clear. The "goal proximity" can be used for LfD as well?

If we consider "goal proximity function" as a goal-related reward function, the method is regarded as the integration of an imitation learning, e.g., GAIL, and a goal-driven reinforcement learning.
From this view, this work looks related to the following paper.

-Kinose, Akira, and Tadahiro Taniguchi. "Integration of imitation learning using GAIL and reinforcement learning using task-achievement rewards via probabilistic graphical model." Advanced Robotics 34.16 (2020): 1055-1067.

---

> ### Author Response · Authors · 2020-11-19
> **Response to Reviewer 3**
>
> We thank the reviewer for the constructive feedback and address the concerns in detail below.
>
> **“This proposal looks like a heuristic method. It is better to clarify the theoretical foundation.”**
> We agree with Reviewer 3 (R3) that our paper proposes a practical solution rather than a theoretically motivated approach. Thus, our paper empirically evaluates the merit of using goal proximity as a means for more robust imitation from observation with exhaustive experiments. While our new reward calculation and the uncertainty penalty are important for our framework, we independently verify how each component contributes to the overall performance. We do so in an updated set of experiments which evaluate the reward formulation (Figure 6), uncertainty penalty (Figure 6), and adversarial training (Figure 5). We then show in what settings our method is advantageous (Figures 4, 8, 9).
>
>
> **“The explicit comparison between the proposed method and GAIL is not given.”**
> As our method is following the form of adversarial imitation learning, our method has a similar structure to GAIL. However, instead of using 1 as the label for expert states in GAIL, our method uses a discounted value for the temporal proximity of that state to the goal. Moreover, the policy learns to maximize the difference in goal proximity every timestep in our method. We elaborated on the comparison between GAIL and our method in Appendix Section B. Moreover, we compared the performance of our method with GAIL and showed comparable performance over most tasks, despite GAIL having access to actions whereas our method does not.
>
>
> **To my understanding, the paper focuses on LfO. However, the relationship between LfO and "goal proximity" is not clear. The "goal proximity" can be used for LfD as well?**
> As R3 mentioned, the concept of goal proximity can work with LfD as well as LfO. However, we found that using actions in the goal proximity hurts generalization performance since the proximity function overfits to the expert actions and hinders sharing proximity information of nearby states. We added this additional experiment in Figure 7a-c.  As we demonstrated throughout our experiments, our method is best when generalizing to unseen states.
>
>
> **Related work [1].**
> We thank R3 for suggesting the relevant paper [1], which incorporates GAIL with the environment reward, and learns to maximize the environment reward as well as the occupancy measure. However, the main idea of our method is to extract a goal-related reward from the expert and agent experiences to replace the occupancy measures from adversarial imitation learning approaches, without access to an environment reward.
>
>
> [1] Kinose and Taniguchi. "Integration of imitation learning using GAIL and reinforcement learning using task-achievement rewards via probabilistic graphical model", Advanced Robotics (2020)

---

### Official Review · AnonReviewer5 · 2020-11-04
**Insufficient evaluation for a paper that comes with no theoratical insights**

**Rating:** 5
**Confidence:** 5

**Review:**

Summary
--------
The paper proposes a method for imitation learning for goal-directed tasks that uses a learned proximity function for computing rewards.
An ensemble of proximity functions is trained in supervised way to predict the time step (rescaled to the range $[0,1]$) for the expert's states. The ensemble is improved online based on the additional objective of assigning zero proximity to the states encountered by the agent. The agent is improved using PPO where the gain in proximity serves as reward function with an additional cost based on the uncertainty about the proximity (estimated by the standard deviation of the ensemble).

Strong points
-------------
- Predicting time from states is an interesting idea and assigning higher reward to later states is sensible for goal directed tasks.
- The paper is well-written and sufficiently clear

Concerns
---------
- Soundness
There is barely any theoretical justification for the approach. Furthermore, several hacks / hyperparameters have been added to achieve the reported results (uncertainty penalty, bonus reward, proximity "discounting", reward scaling).

- Relevance
The approach is limited to goal-directed tasks and thus much less applicable than comparable methods.

- Evaluation
I think that the evaluations are not fair for several reasons.
1. The competitors are not intended for learning under different initial state distributions. It is well known that behavioral cloning often performs very bad for out-of-distribution data. Adversarial methods like GAIL and GAIL-s are based on distribution matching which is in general not possible if the initial state is significantly different from the demonstrations. Especially GAIL-s can suffer here, since it would move to the initial state when starting at the goal position. Matching state transitions as GAILfO would be a bit more reasonable here. Methods that can guide the agent towards the demonstrations, such as SQIL (Reddy et al. 2019) might be even more suitable.

2. The paper mentions that GAIL makes use of additional information by taking into account the actions, but completely ignores the fact that the proposed method makes use of time-labels which are much more relevant for tasks that are essentially defined by the state at the last time step. It would be easy to provide this information also to the adversarial methods by weighting the discriminator samples during training dependent on the time step (in the extrem case by only using the samples form the final time steps). Another naive baseline would be to use a non-parameteric reward function by placing radial basis functions at every expert sample (again weighted dependent on the time step). Instead, the evaluation does not seem to make any use of the strong assumption of goal-directed tasks for any of the competing methods.

3. It seems that much more hyper-parameter tuning was involved for obtaining the results for the proposed method than for competing methods. Even for hyperparameters that are applicable to other methods, such as learning rates and reward scaling the competing methods share the same parameters during all experiments, whereas the proposed method has different hyperparameters depending on the environment.


Additional Feedback
-------------------
Clarity could be improved at some points. Section 3.2. introduces $\delta$ as a discounting but doesn't mention that it is typically set to $1/T$ which essentially scales the predicted time step to the range $[0,1]$. Without this information, it is hard to make sense of Eq. 1 because for a "discount factor" close to 1 (which is typical for the discount factor of the MDP) the proximity function would actually be negative for most time steps and by setting the proximity of agent states to 0 one would actually assign them very high goal proximity.


Questions
---------
1. Why provide an additional reward for last time step? Is this really necessary to obtain good results?

2. I don't see how the optimal proximity function for Eq.4 converges to $\frac{1}{2} - \delta \frac{(T-t)}{2}$. This is not really relevant to the algorithm but still I'm wondering whether this claim is correct. Can you sketch a proof?

3. Can you specify the initial distributions for the test and train experiments more precisely. For example, for the navigation task you mention that 50% of the possible initial states  and goals where used for collecting demonstrations. But, I can neither find the set of possible initial states, nor how the 50% where chosen (uniformly, or selected?).


Assessment
----------
The proposed (heuristic) approach seems reasonable for the considered setting, however, overall the contribution is too small. Strong theoretical results or a thorough empirical evaluation (with suitable baselines) would both be fine for me. However, the current submission lacks quite severly on both sides and is therefore in my opinion not suitable for publication.


References
----------
Reddy, S., Dragan, A. D., Levine, S. SQIL: Imitation Learning via Reinforcement Learning with Sparse Rewards. ICLR. 2019

---

> ### Author Response · Authors · 2020-11-19
> **Response to Reviewer 5**
>
> We thank the reviewer for the constructive feedback and address the concerns in detail below.
>
> **Lack of theoretical justification for approach and several hacks/hyperparameters have been added to achieve the reported results.**
> We agree with Reviewer 5 (R5) that our paper proposes a practical solution rather than a theoretically motivated approach. Thus, our paper empirically evaluates the merit of using goal proximity as a means for robust imitation from observation with exhaustive experiments. While the uncertainty penalty and new reward formulation are important for our method, we verify how each component contributes to the overall performance. We conducted additional experiments which evaluate the reward formulation, uncertainty penalty, and adversarial training in Figure 5 and 6. We then show in what settings our method is advantageous (Figures 4, 8, 9).
>
> Our method introduces two more hyperparameters: the uncertainty penalty weight $\lambda$ and the proximity discounting factor $\delta$. Simply setting $\lambda$ to 0.1 or 0.01 and $\delta$ to around $1/H$ worked across all tasks. Moreover, as shown in Figure 5a and 7d, our method performs similarly well over a range of $\lambda$ and $\delta$ values. Reward scaling can apply to all methods, and we also tried reward scaling for baselines. However, reward scaling was especially beneficial for our method due to the small scale of our difference-based reward, $f(s_{t+1}) - f(s_t)$.
>
>
> **The approach is limited to goal-directed tasks.**
> We agree that the constraint of goal-directed tasks limits the applicability of our method. However, we see goal-driven tasks as an important and broad set of problems, such as object manipulation and navigation. The goal-directed tasks can be also harnessed for higher-level decision making as a sub-skill, such as visual imitation [1].
>
>
> **Comparison to SQIL**
> We agree that SQIL is a suitable baseline. We are currently running experiments on SQIL and will update the paper as soon as we get results.
>
>
> **The evaluation does not seem to make any use of the strong assumption of goal-directed tasks for any of the competing methods.**
> Our core contribution is a method which uses task progress supervision in goal-directed tasks.  In Figure 6, we ablate the uncertainty penalty and our reward formulation from Eq. 2, which is the same as the baseline GAIfO-s with added proximity supervision. Despite this approach using the goal-directed assumption, it only achieves 15.9% more success rate than GAIfO-s, compared to the 43.3% gain of our method. This demonstrates that the combination of proximity, uncertainty, and reward formulation is crucial to our method’s success.
>
>
> **It seems more hyperparameter tuning was involved for the proposed method than competing methods.**
> For our method and baselines, we equally searched over the policy and discriminator learning rates, reward normalization, reward scaling, PPO entropy coefficient, and whether to use state normalization for each task. Our method was more sensitive to the reward scale due to the small scale of our difference-based reward, $f(s_{t+1})-f(s_t)$.
>
>
> **Clarity could be improved about $\delta$ in Section 3.2.**
> We thank R5 for pointing out the clarity of the discounting factor description. As R5 mentioned, $\delta$ scales the goal proximity of states between $[0,1]$ and we thus use a $\delta$ value of around $1/H$ for all experiments. We updated the paper accordingly.
>
>
> **Is the additional reward at the last time step necessary?**
> We conducted additional experiments (Figure 6) and found that the additional reward at the last time step only increases performance by 0.9%, which tells us this additional reward is not a critical component for the improved performance of our method.
>
>
> **“I don't see how the optimal proximity function for Eq. 4 converges to $1/2−\delta(T−t)/2$.”**
> The training objective for the proximity function consists of a loss on agent experience labeled with proximity 0 and another loss on expert experience labeled with proximity  $1-\delta(T-t)$. Since the expert and agent state distributions are the same at optimality, the proximity function will therefore converge to the average of the two labels giving $1/2-\delta(T-t)/2$. This is the same argument as the GAIL discriminator outputting 0.5 at optimality [2].
>
>
> **“Can you specify the initial distributions for the test and train experiments more precisely?”**
> We updated Section 4.3 and 4.4 to clarify the holdout set details. For Navigation, the demonstrations were uniformly randomly sampled from 50% of possible starting states. For 50% on Fetch, the state space was divided into a 4x4 grid, and two cells per row were held out. A similar process is used for the 75% and 25% holdouts.
>
>
> [1] Mandlekar et al., "Learning to Generalize Across Long-Horizon Tasks from Human Demonstrations", RSS 2020
> [2] Fu et al., "Learning robust rewards with adversarial inverse reinforcement learning", ICLR 2018

---

> > ### Author Response · Authors · 2020-11-25
> > **New SQIL Results**
> >
> > As R5 pointed out, adversarial imitation learning baselines are not designed to utilize the goal information and temporal progress, and thus struggle at generalizing to unseen states. This is indeed the contribution of our work: utilizing temporal progress with goal proximity and generalizing to unseen states.
> >
> > As R5 suggested, we compared our method with SQIL [1], which demonstrates high sample efficiency with off-policy RL. The results are added to the paper in Figure 12. Our method consistently outperforms SQIL in the Fetch tasks, despite SQIL using actions whereas our method does not. This can be because the Fetch tasks require precise actions (i.e., it is difficult to reverse moving the block in the wrong direction) which results in covariate shift that negatively impacts BC and SQIL (a form of regularized BC). However, predicting proximity can be easier to generalize and thus provides a useful learning signal on unseen states.
> >
> > On the other hand, SQIL learns the Ant Reach task well and performs similar to the behavioral cloning (BC) baseline since Ant Reach is not sensitive to small errors (i.e., the agent can recover from bad actions and change heading). We observed unstable SQIL results in Figure 12(c), which can also be observed in Figure 2,3 of [1], further training or hyperparameter tuning could stabilize training.
> >
> > We used the same SQIL hyperparameters as [1] and verified our implementation against [1] since there is no publicly available code for SQIL. Due to the limited time and SQIL’s slow training, the training of SQIL has not yet completed 5M environment steps, but we can still observe trends in performance. We will include the complete results in the revised version.
> >
> >
> > [1] Reddy et al., “SQIL: Imitation Learning via Reinforcement Learning with Sparse Rewards”, ICLR 2019

---

### Author Response · Authors · 2020-11-19
**Response to All Reviewers**

We thank all reviewers for the constructive feedback and suggestions. During the rebuttal, we addressed the reviewer’s concerns through (1) exhaustive ablation studies better evaluating our contributions, (2) experiments with a new baseline SQIL, and (3) revised writing clarifying key details of our framework. We believe the revisions significantly improve the empirical evaluation and clarity, and make an important contribution to imitation from observation for goal-directed tasks.

We summarize the major changes to the paper below:

- Performed 5 new ablation experiments for each task analyzing the importance of goal proximity, uncertainty measure, difference-based reward, and their combination in Figure 6. Our new results demonstrate the combination of these three components is critical to the performance of our method. (R1, R2, R3, R4, R5).

- Compared our method to the imitation learning method SQIL in Figure 12. Our method outperforms SQIL in the Fetch tasks even though SQIL learns from actions whereas our method does not (R5).

- Clarified the choice of $\delta$ in Section 3.2 and how our method is robust to different values of $\delta$ in Figure 7d (R1, R5).

- Examined a ranking based loss and action input for the proximity function in Figure 7a-c (R2, R3).

- Updated Section 2 to address additional related work (R2).

- Made an explicit comparison between our method and GAIL in a newly created Appendix Section B (R3).

- Clarified ranking-based proximity training in Section E (R2).

---

### Decision · Program_Chairs · 2021-01-07
**Final Decision**

**Decision:**

Reject

**Comment:**

The paper proposes a method for learning intristic reward from demonstrations. The inartistic reward is computed as time-to-reach and generalizes to unseen states.  The reviewers agree that the method is novel useful, and of interest to ICLR community.

Although the authors' significantly improved the manuscript during the rebuttal phase with new results, and addressed many of the reviewers' comments, the overall novelty of the paper is still somewhat limited, making it unsuitable for ICLR in its current form.

The future  version of the paper should address the comments below and go through a detailed pass for clarity.

Additional comments that did not influence the final decision:

The idea of learning temporal distance to the goal is not novel [1], although the application as an intristic reward is. The authors should connect the temporal difference to the reachability theory and solving two-point boundary problem for systems with non-linear dynamics, as a theoretical foundation of the method [1].

I am curious about the decision to use the time to reach as a reward directly, instead of delta between the states. Some empirical work provides evidence [2,3] that delta yield less side effects in behaviors.

[1] https://ieeexplore.ieee.org/stamp/stamp.jsp?arnumber=8772207
[2]https://arxiv.org/abs/1803.10227
[3] https://arxiv.org/abs/2003.06906